# SAMPLE EFFICIENT ADAPTIVE TEXT-TO-SPEECH

**Yutian Chen, Yannis Assael, Brendan Shillingford, David Budden, Scott Reed,
Heiga Zen, Quan Wang, Luis C. Cobo, Andrew Trask, Ben Laurie,
Caglar Gulcehre, Aäron van den Oord, Oriol Vinyals, Nando de Freitas**

DeepMind & Google
`yutianc@google.com`

## ABSTRACT

We present a meta-learning approach for adaptive text-to-speech (TTS) with few
data. During training, we learn a multi-speaker model using a shared conditional
WaveNet core and independent learned embeddings for each speaker. The aim
of training is not to produce a neural network with fixed weights, which is then
deployed as a TTS system. Instead, the aim is to produce a network that requires
few data at deployment time to rapidly adapt to new speakers. We introduce and
benchmark three strategies: (i) learning the speaker embedding while keeping the
WaveNet core fixed, (ii) fine-tuning the entire architecture with stochastic gradient
descent, and (iii) predicting the speaker embedding with a trained neural network
encoder. The experiments show that these approaches are successful at adapting
the multi-speaker neural network to new speakers, obtaining state-of-the-art results
in both sample naturalness and voice similarity with merely a few minutes of audio
data from new speakers.

## 1 INTRODUCTION

Training a large model with lots of data and subsequently deploying this model to carry out clas-
sification or regression is an important and common methodology in machine learning. It has
been particularly successful in speech recognition (Hinton et al., 2012), machine translation (Wu
et al., 2016) and image recognition (Krizhevsky et al., 2012; Szegedy et al., 2015). In this text-
to-speech (TTS) work, we are instead interested in few-shot meta-learning. Here the objective of
training with many data is not to learn a fixed-parameter classifier, but rather to learn a *"prior" neural
network*. This prior TTS network can be adapted rapidly, using few data, to produce TTS systems for
new speakers at deployment time. That is, the intention is not to learn a fixed final model, but rather
to learn a model prior that harnesses few data at deployment time to learn new behaviours rapidly.
The output of training is not longer a fixed model, but rather a fast learner.

Biology provides motivation for this line of research. It may be argued that evolution is a slow
adaptation process that has resulted in biological machines with the ability to adapt rapidly to new
data during their lifetimes. These machines are born with strong priors that facilitate rapid learning.

We consider a meta-learning approach where the model has two types of parameters: task-dependent
parameters and task-independent parameters. During training, we learn all of these parameters but
discard the task-dependent parameters for deployment. The goal is to use few data to learn the
task-dependent parameters for new tasks rapidly.

Task-dependent parameters play a similar role to latent variables in classical probabilistic graphical
models. Intuitively, these variables introduce flexibility, thus making it easier to learn the task-
independent parameters. For example, in classical HMMs, knowing the latent variables results in a
simple learning problem of estimating the parameters of an exponential-family distribution. In neural
networks, this approach also facilitates learning when there is clear data diversity and categorization.
We show this for adaptive TTS (Dutoit, 1997; Taylor, 2009). In this setting, speakers correspond to
tasks. During training we have many speakers, and it is therefore helpful to have task-dependent
parameters to capture speaker-specific voice styles. At the same time, it is useful to have a large

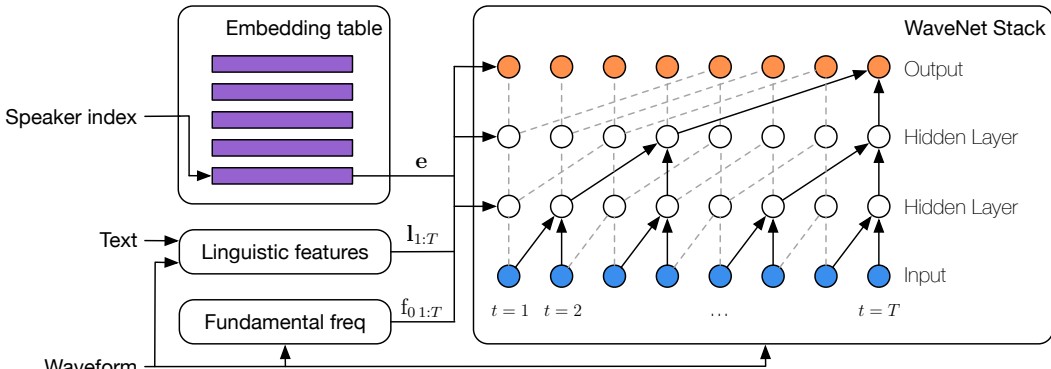

Figure 1: Architecture of the WaveNet model for few-shot voice adaptation.

model with shared parameters to capture the generic process of mapping text to speech. To this end, we employ the WaveNet model.

WaveNet (van den Oord et al., 2016) is an autoregressive generative model for audio waveforms that has yielded state-of-art performance in speech synthesis. This model was later modified for real-time speech generation via probability density distillation into a feed-forward model (van den Oord et al., 2017). A fundamental limitation of WaveNet is the need for hours of training data for each speaker. In this paper we describe a new WaveNet training procedure that facilitates adaptation to new speakers, allowing the synthesis of new voices from no more than 10 minutes of data with high sample quality.

We propose several extensions of WaveNet for sample-efficient adaptive TTS. First, we present two non-parametric adaptation methods that involve fine-tuning either the speaker embeddings only or all the model parameters given few data from a new speaker. Second, we present a parametric text-independent approach whereby an auxiliary network is trained to predict new speaker embeddings.

The experiments will show that all the proposed approaches, when provided with just a few seconds or minutes of recording, can generate high-fidelity utterances that closely resemble the vocal tract characteristics of a demonstration speaker, particularly when the entire model is fine-tuned end-to-end. When fine-tuning by first estimating the speaker embedding and subsequently fine-tuning the entire model, we achieve state-of-the-art results in terms of sample naturalness and voice similarity to target speakers. These results are robust across speech datasets recorded under different conditions and, moreover, we demonstrate that the generated samples are capable of confusing the state-of-the-art text-independent speaker verification system (Wan et al., 2018).

TTS techniques require hours of high-quality recordings, collected in controlled environments, for each new voice style. Given this high cost, reducing the length of the training dataset could be valuable. For example, it is likely to be very beneficial when attempting to restore the voices of patients who suffer from voice-impairing medical conditions. In these cases, long high quality recordings are scarce.

## 2    WAVENET ARCHITECTURE

WaveNet is an autoregressive model that factorizes the joint probability distribution of a waveform, $\mathbf{x} = \{x_1, \ldots, x_T\}$, into a product of conditional distributions using the probabilistic chain rule:

$$p(\mathbf{x}|\mathbf{h}; \mathbf{w}) = \prod_{t=1}^{T} p(x_t|\mathbf{x}_{1:t-1}, \mathbf{h}; \mathbf{w}),$$

where $x_t$ is the $t$-th timestep sample, and $\mathbf{h}$ and $\mathbf{w}$ are respectively the conditioning inputs and parameters of the model. To train a multi-speaker WaveNet, the conditioning inputs $\mathbf{h}$ consist of the speaker identity $s$, the linguistic features $\mathbf{l}$, and the logarithmic fundamental frequency $\mathbf{f}_0$ values. $\mathbf{l}$ encodes the sequence of phonemes derived from the input text, and $\mathbf{f}_0$ controls the dynamics of the pitch in the generated utterance. Given the speaker identity $s$ for each utterance in the dataset, the

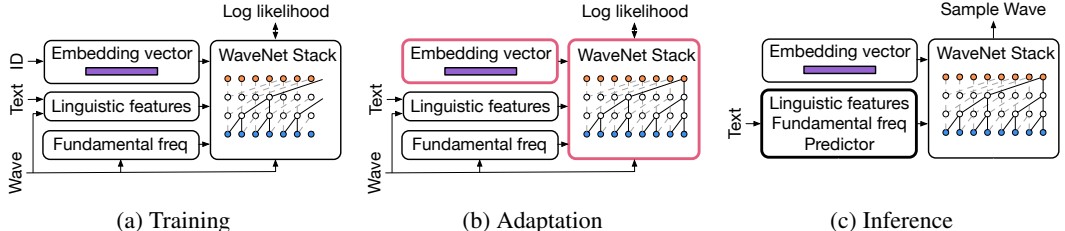

(a) Training  (b) Adaptation  (c) Inference

Figure 2: Training (slow, lots of data), adaptation (fast, few data) and inference stages for the SEA-ALL architecture. The components with bold pink outlines are fine-tuned during the adaptation phase. The purpose of training is to produce a prior. This prior is combined with few data during adaptation to solve a new task. This adapted model is then deployed in the final inference stage.

model is expressed as:

$$p(\mathbf{x}|\mathbf{l}, \mathbf{f}_0; \mathbf{e}_s, \mathbf{w}) = \prod_{t=1}^{T} p(x_t|\mathbf{x}_{1:t-1}, \mathbf{l}, \mathbf{f}_0; \mathbf{e}_s, \mathbf{w}),$$

where a table of speaker embedding vectors $\mathbf{e}_s$ (*Embedding* in Figure 1) is learned alongside the standard WaveNet parameters. These vectors capture salient voice characteristics across individual speakers, and provide a convenient mechanism for generalizing WaveNet to the few-shot adaptation setting in this paper. The linguistic features $\mathbf{l}$ and fundamental frequency values $\mathbf{f}_0$ are both time-series with a lower sampling frequency than the waveform. Thus, to be used as local conditioning variables they are upsampled by a transposed convolutional network. During training, $\mathbf{l}$ and $\mathbf{f}_0$ are extracted by signal processing methods from pairs of training utterance and transcript, and during testing, those values are predicted from text by existing models (Zen et al., 2016).

## 3 FEW-SHOT ADAPTATION WITH WAVENET

In recent years, a large body of literature uses large datasets to train models to learn an input-output mapping that is then used for inference. In contrast, few-shot meta-learning introduces an additional step, adaptation. In this meta-learning setting, the purpose of training becomes to learn a prior. During adaptation, this prior is combined with few data to rapidly learn a new skill; in this case adapting to a new speakers' voice style. Finally, the new skill is deployed, which in this paper we are referring to as inference. These three stages — training, adaptation and inference — are illustrated in Figure 2.

We present two multi-speaker WaveNet extensions for few-shot voice adaptation. First, we introduce a non-parametric model fine-tuning approach, which involves adapting either the speaker embeddings or all the model parameters using held-aside demonstration data. Second, and for comparison purposes, we use a parametric approach whereby an auxiliary network is trained to predict the embedding vector of a new speaker using the demonstration data.

### 3.1 NON-PARAMETRIC FEW-SHOT ADAPTATION VIA FINE-TUNING

Inspired by few-shot learning we first pre-train a multi-speaker conditional WaveNet model on a large and diverse dataset, as described in Section 2. Subsequently, we fine-tune the model parameters by retraining with respect to held-aside adaptation data.

Training this WaveNet model to maximize the conditional log-likelihood of the generated audio jointly optimizes both the set of speaker parameters $\{\mathbf{e}_s\}$ and the shared WaveNet core parameters $\mathbf{w}$. Next, we extend this method to a new speaker by extracting the $\mathbf{l}$ and $\mathbf{f}_0$ features from their adaptation data and randomly initializing a new embedding vector $\mathbf{e}$. We then optimize $\mathbf{e}$ such that the demonstration waveforms, $\{\mathbf{x}_{\text{demo}}^{(1)}, \ldots, \mathbf{x}_{\text{demo}}^{(n)}\}$, paired with features $\{(\mathbf{l}_{\text{demo}}^{(1)}, \mathbf{f}_{0,\text{demo}}^{(1)}), \ldots, (\mathbf{l}_{\text{demo}}^{(n)}, \mathbf{f}_{0,\text{demo}}^{(n)})\}$, are likely under the model with $\mathbf{w}$ fixed (SEA-EMB):

$$\mathbf{e}_{\text{demo}} = \arg\max_{\mathbf{e}} \sum_i \log p(\mathbf{x}_{\text{demo}}^{(i)}|\mathbf{l}_{\text{demo}}^{(i)}, \mathbf{f}_{0,\text{demo}}^{(i)}; \mathbf{e}, \mathbf{w}).$$

Alternatively, all of the model parameters may be additionally fine-tuned (SEA-ALL):

$$(\mathbf{e}_{\text{demo}}, \mathbf{w}_{\text{finetuned}}) = \arg\max_{\mathbf{e}, \mathbf{w}} \sum_i \log p(\mathbf{x}_{\text{demo}}^{(i)} | \mathbf{l}_{\text{demo}}^{(i)}, \mathbf{f}_{0,\text{demo}}^{(i)}; \mathbf{e}, \mathbf{w}).$$

Both methods are non-parametric approaches to few-shot voice adaptation as the number of embedding vectors scales with the number of speakers. However, the training processes are slightly different. Because the SEA-EMB method optimizes only a low-dimensional vector, it is far less prone to overfitting, and we are therefore able to retrain the model to convergence even with mere seconds of adaptation data. By contrast, the SEA-ALL has many more parameters that might overfit to the adaptation data. We therefore hold out $10\%$ of our demonstration data for calculating a standard early termination criterion. We also initialize $\mathbf{e}$ with the optimal value from the SEA-EMB method, and we find this initialization significantly improves the generalization performance even with a few seconds of adaptation data.

### 3.2 PARAMETRIC FEW-SHOT ADAPTATION USING AN EMBEDDING ENCODER

In contrast to the non-parametric approach, whereby a different embedding vector is fitted for each speaker, one can train an auxiliary encoder network to predict an embedding vector for a new speaker given their demonstration data. Specifically, we model:

$$p(\mathbf{x}|\mathbf{l}, \mathbf{f}_0, \mathbf{x}_{\text{demo}}, \mathbf{l}_{\text{demo}}, \mathbf{f}_{0,\text{demo}}; \mathbf{w}) = \prod_{t=1}^{T} p(x_t | \mathbf{x}_{1:t-1}, \mathbf{l}, \mathbf{f}_0; \mathbf{e}(\mathbf{x}_{\text{demo}}, \mathbf{l}_{\text{demo}}, \mathbf{f}_{0,\text{demo}}), \mathbf{w}),$$

where for each training example, we include a randomly selected demonstration utterance from that speaker in addition to the regular conditioning inputs. The full WaveNet model and the encoder network $\mathbf{e}(\cdot)$ are trained together from scratch. We refer the reader to the Appendix for further architectural details. This approach (SEA-ENC) exhibits the advantage of being trained in a transcript-independent setting given only the input waveform, $\mathbf{e}(\mathbf{x}_{\text{demo}})$, and requires negligible computation at adaptation time. However, the learned encoder can also introduce bias when fitting an embedding due to its limited network capacity. As an example, Li et al. (2017) demonstrated a typical scenario whereby speaker identity information can be very quickly extracted with deep models from audio signals. Nonetheless, that the model is less capable of effectively leveraging additional training than approaches based on statistical methods.

### 3.3 REMOVING IDENTITY-RELATED INFORMATION

The linguistic features and fundamental frequencies which are used as inputs contain information specific to an individual speaker. As an example, the average voice pitch in the fundamental frequency sequence is highly speaker-dependent. Instead, we would like these features to be as speaker-independent as possible such that identity is modeled via global conditioning on the speaker embedding. To achieve this, we normalize the fundamental frequency values to have zero mean and unit variance separately for each speaker during training, denoted as $\hat{\mathbf{f}}_0 := (\mathbf{f} - \mathbb{E}[\mathbf{f}_s])/\text{std}(\mathbf{f}_s)$. As mentioned earlier, at test time, we use an existing model (Zen et al., 2016) to predict $(\mathbf{l}, \hat{\mathbf{f}}_0)$.

## 4 RELATED WORK

Few-shot learning to build models, where one can rapidly learn using only a small amount of available data, is one of the most important open challenges in machine learning. Recent studies have attempted to address the problem of few-shot learning by using deep neural networks, and they have shown promising results on classification tasks in vision (Santoro et al., 2016; Shyam et al., 2017) and language (Vinyals et al., 2016). Few-shot learning can also be leveraged in reinforcement learning, such as by imitating human Atari gameplay from a single recorded action sequence (Pohlen et al., 2018) or online video (Aytar et al., 2018).

Meta-learning offers a sound framework for addressing few-shot learning. Here, an expensive learning process results in machines with the ability to learn rapidly from few data. Meta-learning has a long history (Harlow, 1949; Thrun and Pratt, 2012), and recent studies include efforts to learn optimization

processes (Andrychowicz et al., 2016; Chen et al., 2017) that have been shown to extend naturally to the few-shot setting (Ravi and Larochelle, 2016). An alternative approach is model-agnostic meta learning (MAML) (Finn et al., 2017a), which differs by using a fixed optimizer and learning a set of base parameters that can be adapted to minimize any task loss by few steps of gradient descent. This method has shown promise in robotics (Finn et al., 2017b; Yu et al., 2018).

In generative modeling, few-shot learning has been addressed from several perspectives, including matching networks (Bartunov and Vetrov, 2017) and variable inference for memory addressing (Bornschein et al., 2017). Rezende et al. (2016) developed a sequential generative model that extended the Deep Recurrent Attention Writer (DRAW) model (Gregor et al., 2015), and Reed et al. (2018) extended PixelCNN (Van Oord et al., 2016) with neural attention for few-shot auto-regressive density modeling. Veness et al. (2017) presented a gated linear model able to model complex densities from a single pass of a limited dataset.

Early attempts of few-shot adaptation involved the attention models of Reed et al. (2018) and MAML (Finn et al., 2017a), but we found both of these strategies failed to learn informative speaker embedding in our preliminary experiments.

There is growing interest in developing neural TTS models that can be trained end-to-end without the need for hand-crafted representations. In this study we focus on extending the autoregressive WaveNet model (van den Oord et al., 2016; 2017) to the few-shot learning setting to adapt to speakers that were not presented at training time. Other recent neural TTS models include Tacotron 2 (Skerry-Ryan et al., 2018) (building on (Wang et al., 2017)) which uses WaveNet as a vocoder to invert mel-spectrograms generated by an attentive sequence-to-sequence model. DeepVoice 2 (Gibiansky et al., 2017) (building on (Arık et al., 2017)) introduced a multi-speaker variation of Tacotron that learns a low-dimensional embedding for each speaker, which was further extended in DeepVoice 3 (Ping et al., 2018) to a 2,400 multi-speaker scenario. Unlike WaveNet and DeepVoice, the Char2Wav (Sotelo et al., 2017) and VoiceLoop (Taigman et al., 2018) models produce World Vocoder Features (Morise et al., 2016) instead of generating raw audio signals.

Although many of these systems have produced high-quality samples for speakers present in the training set, generalizing to new speakers given only a few seconds of audio remains a challenge. There have been several concurrent works to address this few-shot learning problem. The VoiceLoop model introduced a novel memory-based architecture that was extended by Nachmani et al. (2018) to few-shot voice style adaptation, by introducing an auxiliary fitting network that predicts the embedding of a new speaker. Jia et al. (2018) extended the Tacotron model for one-shot speaker adaptation by conditioning on a speaker embedding vector extracted from a pretrained speaker identity model of Wan et al. (2018). The most similar approached to our work was proposed by Arik et al. (2018) for the DeepVoice 3 model. They considered both predicting the embedding with an encoding network and fitting the embedding based on a small amount of adaptation data, but the adaptation was applied to a prediction model for mel-spectrograms with a fixed vocoder.

## 5 EVALUATION

In this section, we evaluate the quality of samples of SEA-ALL, SEA-EMB and SEA-ENC. We first measure the naturalness of the generated utterances using the standard Mean Opinion Score (MOS) procedure. Then, we evaluate the similarity of generated and real samples using the subjective MOS test and objectively using a speaker verification system (Wan et al., 2018). Finally, we study these results varying the size of the adaptation dataset.

### 5.1 EXPERIMENTAL SETUP

We train a WaveNet model for each of our three methods using the same dataset, which combines the high-quality LibriSpeech audiobook corpus (Panayotov et al., 2015) and a proprietary speech corpus. The LibriSpeech dataset consists of 2302 speakers from the train speaker subsets and approximately 500 hours of utterances, sampled at a frequency of 16 kHz. The proprietary speech corpus consists of 10 American English speakers and approximately 300 hours of utterances, and we down-sample the recording frequency to 16 kHz to match LibriSpeech. The multi-speaker WaveNet model has the same architecture as van den Oord et al. (2016) except that we use a 200-dimensional speaker embedding space to model the large diversity of voices.

| Dataset | LibriSpeech | | VCTK | |
|---|---|---|---|---|
| Real utterance | $4.38 \pm 0.04$ | | $4.45 \pm 0.04$ | |
| van den Oord et al. (2016) | $4.21 \pm 0.081$ | | | |
| Nachmani et al. (2018) | $2.53 \pm 1.11$ | | $3.66 \pm 0.84$ | |
| Arik et al. (2018) | | | | |
|     adapt embedding | - | | $2.67 \pm 0.10$ | |
|     adapt whole-model | - | | $3.16 \pm 0.09$ | |
|     encoding + fine-tuning | - | | $2.99 \pm 0.12$ | |
| Jia et al. (2018) | | | | |
|     trained on LibriSpeech | $\mathbf{4.12 \pm 0.05}$ | | $\mathbf{4.01 \pm 0.06}$ | |
| *Adaptation data size* | *10s* | *<5m* | *10s* | *<10m* |
| SEA-ALL (ours) | $3.94 \pm 0.08$ | $\mathbf{4.13 \pm 0.06}$ | $\mathbf{3.92 \pm 0.07}$ | $\mathbf{3.92 \pm 0.07}$ |
| SEA-EMB (ours) | $3.86 \pm 0.07$ | $3.95 \pm 0.07$ | $3.81 \pm 0.07$ | $3.82 \pm 0.07$ |
| SEA-ENC (ours) | $3.61 \pm 0.06$ | $3.56 \pm 0.06$ | $3.65 \pm 0.06$ | $3.58 \pm 0.06$ |

Table 1: Naturalness of the adapted voices using a 5-scale MOS score (higher is better) with $95\%$ confidence interval on the LibriSpeech and VCTK held-out adaptation datasets. Numbers in bold are the best few-shot learning results on each dataset without statistically significant difference. van den Oord et al. (2016) was trained with 24-hour production quality data, Nachmani et al. (2018) used all samples of each new speaker, Arik et al. (2018) used 10 samples, and Jia et al. (2018) used 5 seconds.

Our few-shot model performance is evaluated using two hold-out datasets. First, the LibriSpeech test corpus consists of 39 speakers, with an average of approximately 52 utterances and 5 minutes of audio per speaker. For every test speaker, we randomly split their demonstration utterances into an adaptation set for adapting our WaveNet models and a test set for evaluation. The subset of utterances used for early termination in Section 3.1 is chosen from the adaptation set. There are about 4.2 utterances on average per speaker in the test set and the rest in the adaptation set. Second, we consider a subset of the CSTR VCTK corpus (Veaux et al., 2017) consisting of 21 American English speakers, with approximately 368 utterances and 12 minutes of audio per speaker. We also apply the adaptation/test split with 10 utterances per speaker for test. We emphasize that no data from VCTK was presented to the model at training time. Since our underlying WaveNet model was trained on data largely from LibriSpeech (which was recorded under noisier conditions than VCTK), one might expect that the generated samples on the VCTK dataset contain characteristic artifacts that make generated samples easier to distinguish from real utterances. However, our evaluation using VCTK indicates that our model generalizes effectively and that such artifacts are not detectable. Synthetic utterances are provided on our demo webpage[1].

It is worth mentioning, that SEA-ENC requires no adaptation time. Where for SEA-EMB, it takes $5 \sim 10k$ optimizing steps to fit the embedding vector, and an additional $100 \sim 200$ steps to fine-tune the entire model using early stopping for SEA-ALL.

## 5.2 NATURALNESS OF THE GENERATED SAMPLES (MOS)

We measure the quality of the generated samples by conducting a MOS test, whereby subjects are asked to rate the naturalness of generated utterances on a five-point Likert Scale (1: Bad, 2: Poor, 3: Fair, 4: Good, 5: Excellent). Furthermore, we compare with other published few-shot TTS systems systems, that were developed in parallel to this work. However, the literature uses varying combinations of training data and evaluation splits making comparison difficult. The results presented are from the closest experimental setups to ours.

Table 1 presents MOS for the adaptation models compared to real utterances. Two different adaptation dataset sizes are considered; $T = 10$ seconds, and $T \leq 5$ minutes for LibriSpeech ($T \leq 10$ minutes for VCTK). For reference on 16 kHz data, WaveNet trained on a 24-hour production quality speech dataset (van den Oord et al., 2016) achieves a score of 4.21, while for LibriSpeech our best few-shot model attains an MOS score of 4.13 using only 5 minutes of data given a pre-trained multi-speaker model. We note that both fine-tuning models produce overall "good" samples for both the LibriSpeech and VCTK test sets, with SEA-ALL outperforming SEA-EMB in all cases. SEA-ALL is on par

---

[1]https://sample-efficient-adaptive-tts.github.io/demo

| Dataset | LibriSpeech | | VCTK | |
|---|---|---|---|---|
| Real utterance | $4.30 \pm 0.08$ | | $4.59 \pm 0.06$ | |
| Jia et al. (2018)
    trained on LibriSpeech | $3.03 \pm 0.09$ | | $2.77 \pm 0.08$ | |
| *Adaptation data size* | *10s* | *<5m* | *10s* | *<10m* |
| SEA-ALL (ours) | $\mathbf{3.41 \pm 0.10}$ | $\mathbf{3.75 \pm 0.09}$ | $\mathbf{3.51 \pm 0.10}$ | $\mathbf{3.97 \pm 0.09}$ |
| SEA-EMB (ours) | $3.42 \pm 0.10$ | $3.56 \pm 0.10$ | $3.07 \pm 0.10$ | $3.18 \pm 0.10$ |
| SEA-ENC (ours) | $2.47 \pm 0.09$ | $2.59 \pm 0.09$ | $2.07 \pm 0.08$ | $2.19 \pm 0.09$ |

Table 2: Voice similarity of generated voices using a 5-scale MOS score (higher is better) with $95\%$ confidence interval on the LibriSpeech and VCTK held-out adaptation datasets.

with the state-of-the-art performance on both datasets. The addition of extra adaptation data beyond 10 seconds of audio helps performance on LibriSpeech but not VCTK, and the gap between our best model and the real utterance is also wider on VCTK, possibly due to the different recording conditions.

## 5.3 VOICE SIMILARITY (MOS)

Beside naturalness, we also measure the similarity of the generated and real voices. The quality of similarity is the main evaluation metric for the voice adaptation problem. We first follow the experiment setup of Jia et al. (2018) to run a MOS test for a subjective assessment and then use a speaker verification model for objective evaluation in the next section.

In every trial of this test a subject is presented with a pair of utterances consisting of a real utterance and another real or generated utterance from the same speaker, and is asked to rate the similarity in voice identity using a five-scale score (1: Not at all similar, 2: Slightly similar, 3: Moderately similar, 4: Very similar, 5: Extremely similar).

Table 2 shows the MOS for real utterances and all the adaptation models under two adaptation data time settings on both datasets. Again, the SEA-ALL model outperforms the other two models, and the improvement over SEA-EMB scales with the amount of adaptation data. Particularly, the learned voices on the VCTK dataset achieve an average score of 3.97, demonstrating the generalization performance on a different dataset. As a rough comparisson, because of varying training setups, the state of the art system of Jia et al. (2018) achieves scores of 3.03 for LibriSpeech and 2.77 for VCTK when trained on LibriSpeech. Their model computes the embedding based on the $d$-vector, similar to our SEA-ENC approach, and performs competitively for the one-shot learning setting, but its performance saturates with 5 seconds of adaptation data, as explained in Section 3.2. We note the gap of similarity scores between SEA-ALL and real utterances, which suggests that although the generated samples sound similar to the target speakers, humans can still tell the difference from real utterances.

## 5.4 VOICE SIMILARITY (SPEAKER VERIFICATION)

We also apply the state-of-the-art text independent speaker verification (TI-SV) model of (Wan et al., 2018) to objectively assess whether the generated samples preserve the acoustic features of the speakers. We calculate the TI-SV $d$-vector embeddings for generated and real voices. In Figure 3, we visualize the 2-dimensional projection of the $d$-vectors for a SEA-ALL model trained on $T \leq 5$ minutes of data on the LibriSpeech dataset, and $T \leq 10$ minutes on VCTK. There are clear clusters on both datasets, with a strikingly large inter-cluster distance and low intra-cluster separation. This shows both (1) an ease of correctly identifying the speaker associated with a given generated utterance, and (2) the difficulty in differentiating real from synthetic samples.

A similar figure is presented in (Jia et al., 2018), but there the generated and real samples do not overlap. This indicates that the method presented in this paper generates voices that are more indistinguishable from real ones, when measured with the same verification system. In the following subsections, we further analyze these results.

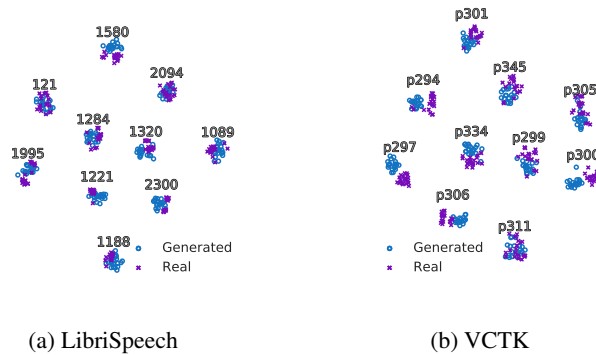

| (a) LibriSpeech | (b) VCTK |

Figure 3: t-SNE visualization of the $d$-vector embeddings of real and SEA-ALL-generated utterances, for both the LibriSpeech ($T \leq 5$ mins) and VCTK ($T \leq 10$ mins) evaluation datasets.

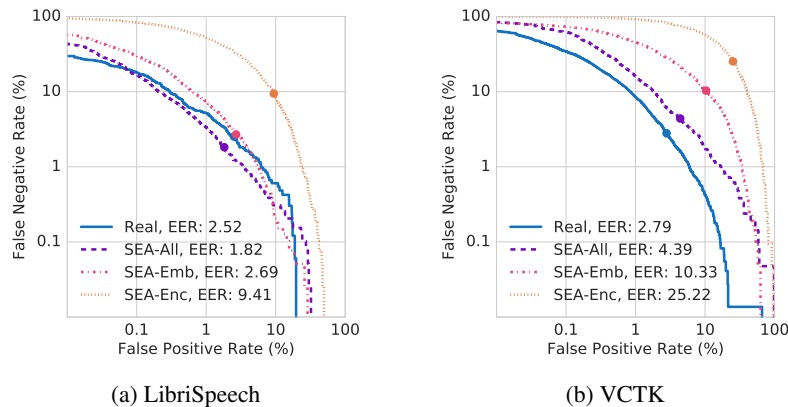

| (a) LibriSpeech | (b) VCTK |

Figure 4: Detection error trade-off (DET) curve for speaker verification in percentage, using the TI-SV speaker verification model (Wan et al., 2018). The utterances were generated using $T \leq 5$ and $T \leq 10$ minute samples from LibriSpeech and VCTK respectively. EER is marked with a dot.

### 5.4.1 DISCERNING DIFFERENT SPEAKERS

We first quantify whether generated utterances are attributed to the correct speaker. Following common practice in speaker verification (Wan et al., 2018), we select the hold-out test set of real utterances from test speakers as the enrollment set and compute the centroid of the $d$-vectors for each speaker $c_i$. We then use the adaptation set of test speakers as the verification set. For every verification utterance, we compute the cosine similarity between its $d$-vector $\mathbf{v}$ and a *randomly* chosen centroid $c_i$. The utterance is accepted as one from speaker $i$ if the similarity is exceeds a given threshold. We repeat the experiments with the same enrollment set and replace the verification set with samples generated by each adaptation method under different data size settings.

| Dataset | LibriSpeech | | | VCTK | | |
|---|---|---|---|---|---|---|
| Real utterance | 2.47 | | | 2.79 | | |
| *Adaptation data size* | *10s* | *1m* | *<5m* | *10s* | *1m* | *<10m* |
| SEA-ALL (ours) | **3.17** | **2.47** | **1.85** | **7.34** | **5.02** | **4.33** |
| SEA-EMB (ours) | 3.26 | 2.92 | 2.74 | 10.18 | 9.91 | 10.24 |
| SEA-ENC (ours) | 10.73 | 9.77 | 9.42 | 27.20 | 25.34 | 25.23 |

Table 3: Equal error rate (EER) of real and few-shot adapted voice samples for evaluation of voice similarity. Varying adaptation dataset sizes were considered.

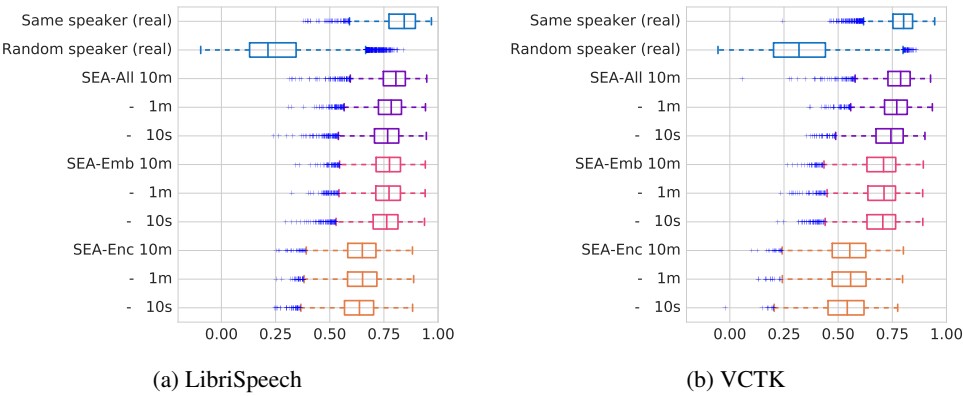

(a) LibriSpeech

(b) VCTK

Figure 5: Cosine similarity of real and generated utterances to the real enrollment set.

In our setup we fix the enrollment set together with the speaker verification model from (Wan et al., 2018), and study the performance of different verification sets that are either from real utterances or generated by a TTS system. Table 3 lists the equal error rate (EER) of the verification model with real and generated verification utterances, and Figure 4 shows the detection error trade-off (DET) curves for a more thorough inspection. Figure 4 only shows the adaptation models with the maximum data size setting ($T \leq 5$ minutes for LibriSpeech and $\leq 10$ minutes for VCTK). The results for other data sizes are provided in Appendix B.

We find that SEA-ALL outperforms the other two approaches, and the error rate decreases clearly with the size of demonstration data. Noticeably, the EER of SEA-ALL is even lower than the real utterance on the LibriSpeech dataset with sufficient adaptation data. A possible explanation is that the generated samples might be concentrated closer to the centroid of a speaker's embeddings than real speech with larger variance across utterances. Our SEA-EMB model performs better than SEA-ENC. Additionally, the benefit of more demonstration data is less significant than for SEA-ALL in both of these models.

### 5.4.2 DISCERNING REAL FROM GENERATED UTTERANCES

In this section, we compare the generated samples and the real utterances of the speaker being imitated. Figure 5 shows the box-plot of the cosine similarity between the embedding centroids of test speakers' enrollment set and (1) real utterances from the same speaker, (2) real utterances from a different speaker, and (3) generated utterances adapted to the same speaker. Consistent with the observations from the previous subsection, SEA-ALL performs best.

We further consider an adversarial scenario for speaker verification. In contrast to the previous standard speaker verification setup where we now select a verification utterance with either a real utterance from the *same* speaker or a synthetic sample from a model adapted to the *same* speaker. Under this setup, the speaker verification system is challenged by synthetic samples and acts as a classifier for real versus generated utterances. The ROC curve of this setup is shown in Figure 6 and the models are using the maximum data size setting. Other data size settings can be found in Appendix C. If the generated samples are indistinguishable from real utterances, the ROC curve approaches the diagonal line (that is, the verification system fails to separate real and generated voices). Importantly, SEA-ALL manages to confuse the verification system especially for the VCTK dataset where the ROC curve is almost inline with the diagonal line with an AUC of $0.56$.

## 6 CONCLUSION

This paper studied three variants of meta-learning for sample efficient adaptive TTS. The adaptation method that fine-tunes the entire model, with the speaker embedding vector first optimized, shows impressive performance even with only 10 seconds of audio from new speakers. When adapted with a few minutes of data, our model matches the state-of-the-art performance in sample naturalness. Moreover, it outperforms other recent works in matching the new speaker's voice. We also demon-

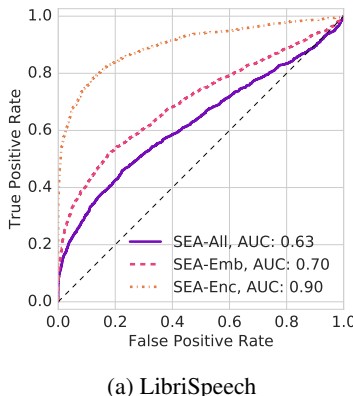
(a) LibriSpeech

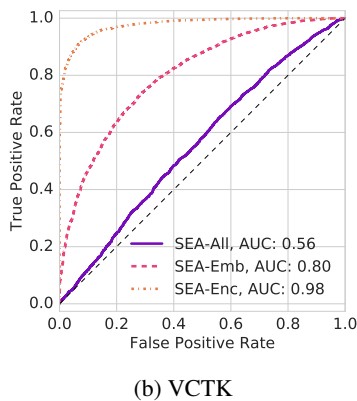
(b) VCTK

Figure 6: ROC curve for real versus generated utterance detection. The utterances were generated using models with 5 and 10 minutes of training data per speaker from LibriSpeech and VCTK respectively. Lower curve indicate that the verification system is having a harder time distinguishing real from generated samples.

strated that the generated samples achieved a similar level of voice similarity to real utterances from the same speaker, when measured by a text independent speaker verification model.

Our paper considers the adaptation to new voices with clean, high-quality training data collected in a controlled environment. The few-shot learning of voices with noisy data is beyond the scope of this paper and remains a challenging open research problem.

A requirement for less training data to adapt the model, however, increases the potential for both beneficial and harmful applications of text-to-speech technologies such as the creation of synthesized media. While the requirements for this particular model (including the high-quality training data collected in a controlled environment and equally high quality data from the speakers to which we adapt, as described in Section 5.1) present barriers to misuse, more research must be conducted to mitigate and detect instances of misuse of text-to-speech technologies in general.

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

## A    EMBEDDING ENCODER

Our encoding network is illustrated as the summation of two sub-network outputs in Figure 7. The first sub-network is a pre-trained speaker verification model (TI-SV) (Wan et al., 2018), comprising 3 LSTM layers and a single linear layer. This model maps a waveform sequence of arbitrary length to a fixed 256-dimensional $d$-vector with a sliding window, and is trained from approximately 36M utterances from 18K speakers extracted from anonymized voice search logs. On top of this we add a shallow MLP to project the output $d$-vector to the speaker embedding space. The second sub-network comprises 16 1-D convolutional layers. This network reduces the temporal resolution to 256 ms per frame (for 16 kHz audio), then averages across time and projects into the speaker embedding space. The purpose of this network is to extract residual speaker information present in the demonstration waveforms but not captured by the pre-trained TI-SV model.

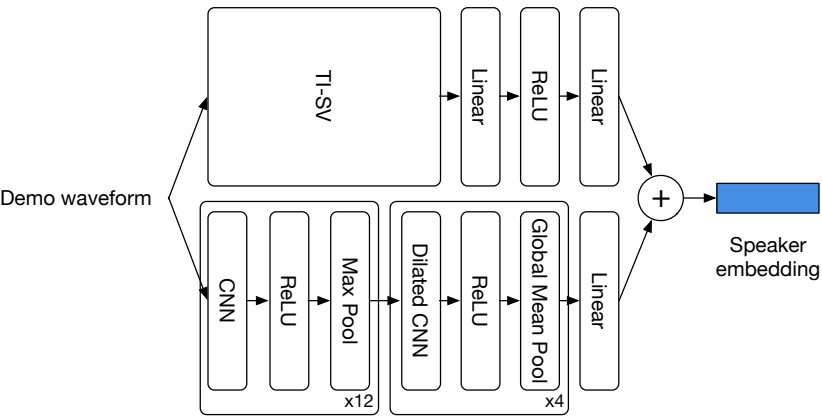

Figure 7: Encoder network architecture for predicting speaker embeddings.

## B DET CURVES VARYING TRAINING DATA SIZES

Here we provide the DET curves of speaker verification problem for models with different training data sizes in addition to those shown in Section 5.4.1.

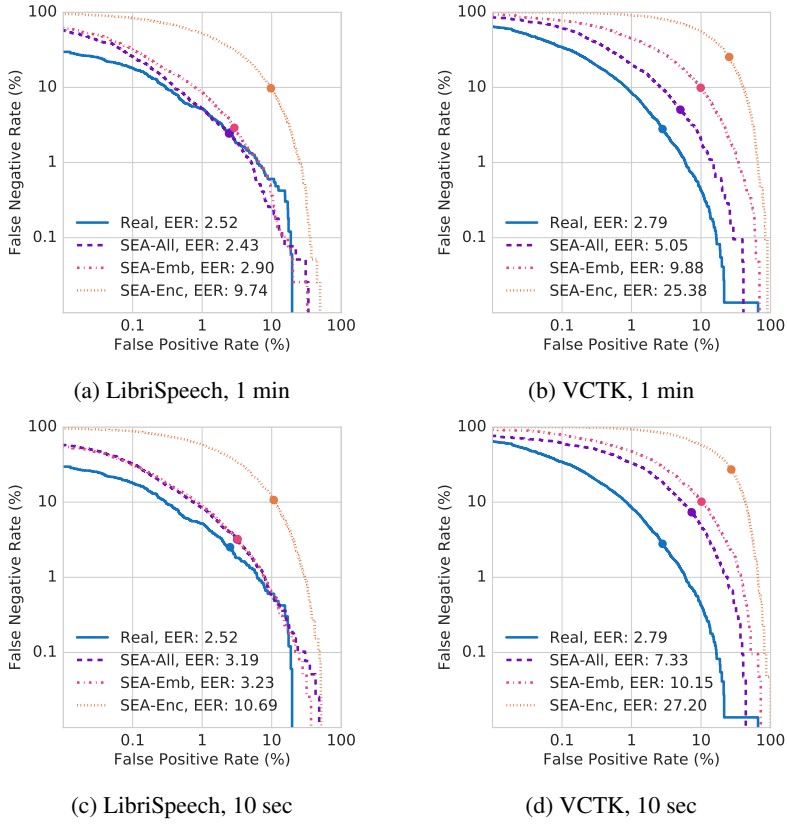

Figure 8: Detection error trade-off (DET) curve for speaker verification, using the TI-SV speaker verification model (Wan et al., 2018). The utterances were generated using 1 minute or 10 seconds of utterance from LibriSpeech and VCTK. EER is marked with a dot.

## C  ROC CURVES VARYING TRAINING DATA SIZES

We provide the ROC curves of the speaker verification problem with adversarial examples from adaptation models with different training data sizes in addition to those shown in Section 5.4.2.

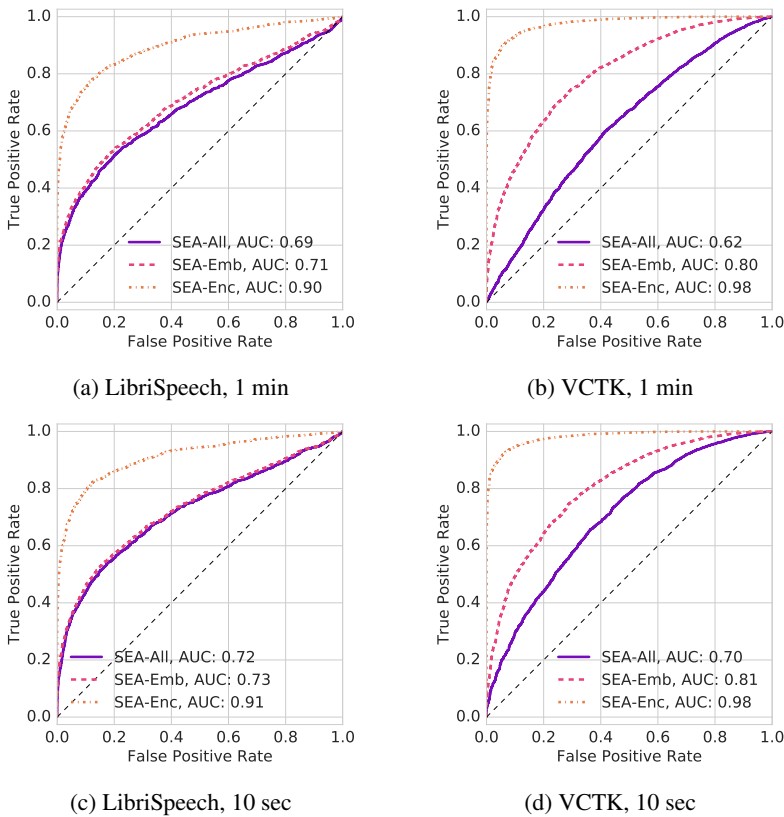

Figure 9: ROC curve for real vs. generated utterance detection. The utterances were generated using 1 minute or 10 seconds of utterance from LibriSpeech and VCTK. Lower curve suggests harder to distinguish real from generated samples.

