# OpenReview forum: "Sample Efficient Adaptive Text-to-Speech"
_ICLR.cc/2019/Conference_

### Official Review · AnonReviewer3 · 2018-11-01
**A good work; limited novelty but solid results**

**Rating:** 6
**Confidence:** 5

**Review:**

This paper investigates speaker adaption with a few samples based on an existing (pre-trained) multi-speaker TTS system. The three approaches in this paper are almost the same as the voice cloning work in Arik et al. (2018). However, it is still very beneficial to demonstrate these approaches for linguistic feature conditioned WaveNet.

Detailed comments:

1) This manuscript is not self-contained, as it omits the important details for acquiring linguistic features (e.g., phoneme duration model) and fundamental frequency (F0) at training and test time. The only information is that it uses existing model (Zen et al., 2016) to predict linguistic features and F0. What type of linguistic features are used in this work? Is the existing model (Zen et al., 2016) trained on the same training set as WaveNet model?

2) It seems the only speaker-dependent part of the system is the embedding table for WaveNet. Actually, both linguistic features (e.g., phoneme duration) and fundamental frequency sequence are highly speaker-dependent. The authors normalize F0 to make it as speaker-independent as possible. What about the speaker-dependent linguistic features? Why not keep them as speaker dependent, and do speaker-adaption for the new speaker at inference?

3) In my opinion, it’s a bit superfluous to name fine tuning as non-parametric few-shot adaption, and auxiliary network (speaker encoding) as parametric few-short adaption. Both ideas are quite natural as in Arik et al. (2018).

4) The abbreviations SEA-ALL, SEA-EMB and SEA-ENC are appeared without explanation.

5) It would be better to provide more details about early termination criterion in Section 3.1. Is it simply the validation loss?

6) In Table 1, the MOS from Arik et al. (2018) and Jia et al. (2018) are not comparable. The experimental settings are different. Perhaps more importantly, these MOS evaluations are done by different group of people.

7) In Section 5.3, Nachmani et al. (2018) and Arik et al. (2018) have also used speaker verification model as an objective evaluation.

Overall, this is a good work with limited novelty but solid results. However, it can be improved in many ways as detailed  in previous comments. I would like to raise my rating if these comments can be addressed properly.

---

> ### Author Response · Authors · 2018-11-15
> **Thank you for your comments. A revision is submitted following the feedback from all reviewers.**
>
> Thanks for your insightful comments. As discussed in the related work section, our proposed approaches are closely related to the methods in Arik et al. (2018) at a high level. However, this is a situation where the details seem to matter significantly. For instance, we find that the detail of applying few-shot adaptation to the WaveNet core results in better sample quality.
>
> We have submitted a revision incorporating all reviewers’ comments. Please see our response to your other questions below and let us know if any explanation is unclear.
>
> 1) Details for acquiring linguistic features:
> We did not provide a fine-grained explanation of the linguistic features originally because we used the same pipeline as van den Oord et al., (2016). We have now added a new section in the supplementary material of the revised paper (Section A) to explain how to extract the linguistic features and fundamental frequency at training and adaptation phases, and how to predict both at the inference phase.
>
> 2) Speaker-dependent linguistic features
> The linguistic features and F0 are provided as inputs during training, but are not available at inference time. Our experimental evaluation shows that even though the linguistic features are predicted by a standard speaker model, the voice similarity is high. This is because the speaker identity is closely dependent to the vocal tract properties, which are modelled by WaveNet. Nevertheless, we strongly agree that the linguistic features also contribute to the speaker identity in terms of prosody, accent, and so on. For this reason, we are considering future work on using our few-shot method for adapting F0 and linguistic features in the hope of improving performance.
>
> 3) Superfluous naming
> You raise a valid point. We however felt that those names captured the difference between the two approaches clearly, and fitted well with the overall presentation. If you think they reduce readability, please let us know and we’ll consider alternatives.
>
> 4) Method abbreviations without explanation
> SEA stands for our “sample efficient adaptive” method as in the title. We have clarified this in the revision.
>
> 5) Early stopping criterion
> The early stopping criterion uses the validation loss on the hold-out data.
>
> 6) Not comparable MOS in Table 1.
> You’re absolutely right. As discussed in Section 5.2, comparing all other methods in the exact same setting is challenging as it would require that we retrain all models on the same dataset with hyper-parameter sweeps. We did our best to make the setups as close as possible when reporting our MOS results and results from other references. Please note that the same five-point Likert Scale is shared by all evaluations, and the group of human raters is shared with Jia et al. (2018).
>
> 7) Nachmani et al. (2018) and Arik et al. (2018) have also used speaker verification model as an objective evaluation.
> This is true indeed. We even considered using this to compare approaches. However, any comparison with a speaker verification model is highly dependent on the model architecture, the training set, and the set of test speakers (both the identity and the size of the test set). For this reason, we decided to not include a naive comparison, even though we did find that we get the lowest equal error rate with SEA-ALL.

---

### Official Review · AnonReviewer1 · 2018-11-03
**Wavenet solution to speaker adaptation - accept**

**Rating:** 7
**Confidence:** 4

**Review:**

This paper proposes  an adaptation technique for TTS using wavenet as the speech backend, with the adaptation carried out on small data. The work is extremely significant in that speech data is hard to produce  (we need many hours of speaker data), and techniques to adapt (transfer learning?) data from large networks would be quite valuable. The main idea is that we train a network containing a large amount of data, and (assuming that we have a trained model), we adapt this network to the task of generating speech from text for a much smaller dataset.

In general (insofar as we can use that term), one trains such a network using <text/speech> pairs,  with speaker conditioning as added input so as to produce voice from a given speaker. The input text is converted to linguistic features in the ‘front end’, which is then injected with voice features to be synthesized into a voice output in the backend. More recent efforts in speech modeling have used RNN or wavenet based systems to carry out these transformations in the front/backends. The present work seems to use an SPSS technique (Zen et al 2016) to generate the linguistic features, while the task of converting to voice is carried out by a Wavenet.

The work is quite (conceptually) similar to "Neural voice cloning with a few samples" (Arik et al, https://arxiv.org/abs/1802.06006) in proposing techniques for few shot adaptation described below but with the significant difference that the latter used autoregressive DNNs (loosely speaking, seq2seq a la Tacotron) for the task in both the front and back ends, while in the current work, the linguistic features are computed with SPSS as in Zel et al (2016).

The paper proposes three quite related techniques for adaptation as shown clearly in Figure 2 of the paper. These techniques are again ‘roughly’ analogous to those described in the Baidu work “Neural Voice Cloning with a few samples”, with the difference in front and backend setups noted in the previous paragraph.

We take as text as input, and convert them to a representation for linguistic features as described in Zen et al (2016). To this, we now add the fundamental frequency F_0 for the sample voice. The key piece needed is the speaker embeddings (a vector), which is to be obtained by training. In addition to all this, we also have available the weights of a trained wavenet network (probably quite large) trained on many speakers, which we will modify (or not) using the strategies outlined for the few data dataset.

SEA-EMB - Train embeddings, but not the network. We expect this to be ‘fast’, but not particularly accurate.
SEA-ALL - Train embeddings, and network. This would be a much more accurate, if slower task. The authors note that since we train a very large network in this case, it could be prone to overfitting. They employ early stopping (as a practitioner, I would make note of the issue) with 10 % of the dataset being held out. Additional ideas such as initializing the emeddings - possibly with those that SEA-EMB calculates - are also stated to be useful.
SEA-ENC - In this third version, they predict speaker embeddings from the trained larger network (the recipe is provided in the appendix). This task of predicting speaker embeddings is one of training a classifier.


Results
The paper presents evaluations conducted with subjective, MOS based enrolment and with an evaluation metric from TI-SV d-vectors. Comparisons are made for all three models with human evaluated MOS scores, and it is seen that SEA-ALL outperforms the other two models, while performance in SEA-EMB depends on the amount of data used. Nevertheless, humans are still able to detect the difference between synthetic voices and real samples.

The TI-SV evaluations from Wan et al show t-SNE embeddings of ‘clusters’ of d-vectors for human and synthetic voices, where it is seen that inter-cluster distance (i.e. between different speakers) is high, showing that the model is able to discern speakers, and the intra-cluster distance (i.e. between real and synthetic voices) is low, showing that synthetic voices are ‘similar’ to real voices. In addition, three other measures - cosine similarity, and statistical measures for detection error trade off, ROC curves and cosine similarity measures are also presented, which show that that the adaptation models perform quite well.


Clarifications and comments:

Have there been efforts to compare this model (with the SPSS based frontend) with seq2seq (Bahdanau/transformer) DNN based systems as in “Neural Voice cloning with few samples”?. How do they compare (is it even a valid comparison?)?

I think the model for computing linguistic features could be elaborated upon further.

Representations: I assume that the output audio representation is an audio waveform

Typo 1 (minor): The reference  for “Bornschein et al” in section 4 “Related work”
“Variable inference for memory addressing”.
Correction “Variational memory addressing in generative models”

Typo 2 (minor): Figure 6: Lower curve indicate that the verification system is having a harder time distinguishing real from generated samples.
Correction (minor): Lower curve “indicates” ...

Summary
-------------
In summary, I am in favor of accepting this paper as it proposes a solution to adapt a trained network to one with has limited number of samples. A big issue in speech modeling is that datasets are tiny, and it is difficult to obtain good quality data at reasonable cost. It would be extremely useful to have a trained network that we can adapt for our own experiments. The related paper by Arik et al (Neural Voice cloning with a few samples) also operates with similar strategic aims, but uses a a different methodology using attention based DNNs. The paper under review should be a good addition to the toolbox of few shot adaptation/transfer learning for speech with much potential for practical use.

---

> ### Author Response · Authors · 2018-11-15
> **Thank you for your comments. A revision is submitted following the feedback from all reviewers.**
>
> Thanks a lot for your terrific summary of our paper. We’ve submitted a revision following the feedback from all reviewers. Please see our response to your questions below:
>
> - Comparison between SPSS based frontend and seq2seq models:
> There are pros and cons to both approaches. On the one hand, SPSS+WaveNet obtains a natural decomposition of prosody (pace, intonation, etc) and vocal tract properties (more relevant to speaker identity) of a voice, something that is still difficult to do with seq2seq [1]. On the other hand, seq2seq models overcome the need of hand-crafted linguistic features and could be easily applied to different languages.
>
> We compare the performance of our model with seq2seq models in terms of sample naturalness and voice similarity in Tables 1 and 2. However, as explained in our paper, we report the numbers of the closest experimental setup. Without access to the original code and all the dataset-specific hyper-parameters it is difficult to reproduce other works exactly.
>
> Listening to the generated samples on our demo webpage is another way to qualitatively compare the approaches.
>
> - Linguistic features:
> Please refer to our response to question 1 of Reviewer 3 for details. In short, we have added an additional appendix to the paper elaborating on the linguistic features.
>
> - Representations:
> Correct, the output is an audio waveform.
>
> Thanks for pointing out the typos.
>
> References:
> [1]: Skerry-Ryan, R. J., et al. "Towards End-to-End Prosody Transfer for Expressive Speech Synthesis with Tacotron." arXiv preprint arXiv:1803.09047 (2018).

---

### Official Review · AnonReviewer2 · 2018-11-06
**nice paper, publish**

**Rating:** 7
**Confidence:** 4

**Review:**

This paper presents an approach to customize or adapt a text-to-speech synthesis system to a new speaker, given relatively small amount of data from that speaker.  It is a very well written paper with rather strong results indicating high quality, naturalness, and similarity with real speech from a speaker can be achieved with the authors' proposed approach.  I think the paper should be accepted for presentation at the conference.

Few comments:
a) In second equation in Section 2 authors state speaker identity “s” is part of conditioning inputs “h” but it is not shown in the Equation where “h” is replaced with “l, f_0”
b) In related work, I think the speaker code work of Abdel-Hamid et al., e.g. Ossama Abdel-Hamid, Hui Jiang, “Fast speaker adaptation of hybrid NN/HMM model for speech recognition based on discriminative learning of speaker code,” ICASSP 2013 is worth citing.
c) The result that synthesized speech performs better than real speech in speaker verification task is interesting.  To me this points to a potential weakness in the verification methodology.  Please comment if this may be the case.

---

> ### Author Response · Authors · 2018-11-15
> **Thank you for your comments. A revision is submitted following the feedback from all reviewers.**
>
> Thank you for your supportive comments. We’ve submitted a revision following the feedback from all reviewers. Please see our response to your questions below:
>
> a) Conditioning inputs:
> Speaker identity is part of h. Because s is used to select the speaker-specific embedding parameter, we include it in the second equation as a subscript in e_s. We have added a note to clarify this in the revision.
>
> b) Citation to Fast speaker adaptation for speech recognition:
> Thanks for bringing this paper to our attention. We’ve included it in the revision.
>
> c) Synthesized speech outperforms real speech in speaker verification task:
> This is a good point. Our experiments suggest that synthesized samples from SEA-ALL on LibriSpeech deviate less from the *centroid* of real utterances than the real samples. Ideally, when comparing samples of different generative models to real utterances we would like these to match in distribution and not only in terms of scalar point estimates. In our practical setting, we would like the generated and real samples to have overlapping tSNE projections as in Figure 3, and to have similar DET curves as in Figure 4. We expanded on this point in the revised version of the paper.

---

### Meta-Review · Area_Chair1 · 2018-12-14
**Limited novelty but sound experimental work**

**Confidence:** 4
**Recommendation:** Accept (Poster)

**Metareview:**

The paper benchmarks three strategies to adapt an existing TTS system (based on WaveNet) to new speakers.

The paper is clearly written. The models and adaptation strategies are not very novel, but still a scientific contribution. Overall, the experimental results are detailed and convincing. The rebuttals addressed some of the concerns.

This is a welcomed contribution to ICLR 2019.